# Image Fusion Involving Real-Time Transabdominal or Endoscopic Ultrasound for Gastrointestinal Malignancies: Review of Current and Future Applications

**DOI:** 10.3390/diagnostics12123218

**Published:** 2022-12-19

**Authors:** Ben S. Singh, Irina M. Cazacu, Carlos A. Deza, Bastien S. Rigaud, Adrian Saftoiu, Gabriel Gruionu, Lucian Guionu, Kristy K. Brock, Eugene J. Koay, Joseph M. Herman, Manoop S. Bhutani

**Affiliations:** 1Department of Gastroenterology, Hepatology and Nutrition, The University of Texas MD Anderson Cancer Center, Houston, TX 77030, USA; 2Department of Oncology, Fundeni Clinical Institute, 022328 Bucharest, Romania; 3Faculty of Medicine, University of Medicine and Pharmacy Carol Davila, 050474 Bucharest, Romania; 4Department of Radiation Oncology, The University of Texas MD Anderson Cancer Center, Houston, TX 77030, USA; 5Department of Imaging Physics, The University of Texas MD Anderson Cancer Center, Houston, TX 77030, USA; 6Department of Gastroenterology and Hepatology, Elias Emergency University Hospital, 011461 Bucharest, Romania; 7Department of Gastroenterology, Ponderas Academic Hospital, 014142 Bucharest, Romania; 8Department of Medicine, Indiana University School of Medicine, Indianapolis, IN 46202, USA; 9Department of Mechanics, University of Craiova, 200585 Craiova, Romania; 10Department of Radiation Medicine, Zucher School of Medicine, Hempstead, NY 11549, USA

**Keywords:** image fusion, transabdominal ultrasound, endoscopic ultrasound, GI malignancies

## Abstract

Image fusion of CT, MRI, and PET with endoscopic ultrasound and transabdominal ultrasound can be promising for GI malignancies as it has the potential to allow for a more precise lesion characterization with higher accuracy in tumor detection, staging, and interventional/image guidance. We conducted a literature review to identify the current possibilities of real-time image fusion involving US with a focus on clinical applications in the management of GI malignancies. Liver applications have been the most extensively investigated, either in experimental or commercially available systems. Real-time US fusion imaging of the liver is gaining more acceptance as it enables further diagnosis and interventional therapy of focal liver lesions that are difficult to visualize using conventional B-mode ultrasound. Clinical studies on EUS guided image fusion, to date, are limited. EUS–CT image fusion allowed for easier navigation and profiling of the target tumor and/or surrounding anatomical structure. Image fusion techniques encompassing multiple imaging modalities appear to be feasible and have been observed to increase visualization accuracy during interventional and diagnostic applications.

## 1. Introduction

The accurate diagnosis and staging of gastrointestinal malignancies can be further enhanced through the use of image fusion involving ultrasound (US). Medical image fusion refers to the integration and merging of visual information from various imaging modalities. Image fusion involving US incorporates the dynamic imaging technique of transabdominal ultrasound (TUS) or endoscopic ultrasound (EUS) fused with computed tomography (CT), magnetic resonance imaging (MRI), or positron emission tomography (PET). The major advantages of combining various imaging techniques include the ability to compare findings from one modality to another and the overall improved visualization of target lesions for diagnosis and image-guided interventions. The fusion of these modalities for clinical use has increased in recent years and has extended the diagnostic and therapeutic capabilities of US, enabling physicians to have a more precise assessment of target lesions [1,2,3,4]. For example, contrast-enhanced ultrasound (CE-US) fusion with helical CT or MRI has been reported to show a tendency for increased detection of liver lesions when compared with CE-US alone [5]. Similarly, fusion of EUS with CT was seen to be accurate and increase detection and characterization of pancreato-biliary and mediastinal lesions [2].

Image fusion of CT, MRI, and PET with US can be promising for GI malignancies as it has the potential to allow for a more precise lesion characterization with higher accuracy in tumor detection, staging, and interventional/image guidance. Here, we conducted a literature review to present an overview of the current possibilities of real-time image fusion involving US with a focus on clinical applications in the management of GI malignancies.

## 2. Methods

The literature search was performed in PubMed and Web of Science using the terms “ultrasonography” or “endoscopic ultrasound” and “image fusion” and “gastrointestinal” or “digestive” and “cancer” or “malignancies”. The reference lists of the retrieved articles were hand-searched for further references.

## 3. Image Modalities for Fusion and Benefits of Image Fusion

Precise image guidance is necessary for managing GI malignancies but can be challenging due to the possible small size of tumors, overall poor visualization, and deformable nature of the surrounding tissue.

CT imaging is one of the main pillars of image-guided procedures for GI tumors due to its high spatial resolution and the ability to image all vital structures. However, it is limited by the lack of real-time feedback, radiation exposure, and need for contrast for tumor visualization [6]. 

Similarly, MRI is a common diagnostic imaging modality for organs such as the liver or pancreas due to its ability to obtain superior tissue contrast resolution and high-resolution images without ionizing radiation. Real-time MRI guidance is available, but it requires a prolonged procedure time; the images are susceptible to motion artifact (especially with breathing motion) and the additional equipment is expensive and cumbersome. MRI also poses safety concerns for patients with contraindications such as the presence of electronic medical devices or metal implants.

On the other hand, US is widely available, portable, and less costly. Therefore, improvement in US technology and utilization through the integration of image fusion could have a global impact on improving the quality of care of patients with GI malignancies. US allows real-time visualization and does not have the same issues of radiation exposure, as with CT, or the safety risks and training required, as with MRI. Therefore, one can appreciate the advantage of US guidance to target GI tumors and adjacent tissues by fusing diagnostic quality MRI/CT images. With image fusion, the practicability of real-time US guidance is combined with the high resolution of baseline CT/MRI datasets. Furthermore, image fusion incorporating endoscopic US (EUS) with CT/MRI imaging can provide real-time, high-resolution visualization of liver lesions and pancreato-biliary and esophageal lesions through multiple viewing planes [1,6]. EUS image fusion has the potential to increase the accuracy of EUS-guided interventions and tumor staging [7,8]. The combination of dynamic images from EUS and baseline CT images can help establish an enhanced radiation, surgical, and interventional treatment plan through the greater profiling of the pancreatic tumor while assessing the location of surrounding tissues and vascular involvement by the target lesion [9].

Perhaps the most well-known fusion application in diagnostic imaging is the use of functional FDG-PET with CT for anatomic localization. PET/CT is a powerful tool for the assessment of metabolic activity of lesions suspected of malignancy and whole-body oncologic staging [10,11]. However, cost, lack of reimbursement, radiation exposure, and the prolonged acquisition time limit the widespread use of PET/CT-guided interventions. Fusion of pre-acquired PET/CT data to intraprocedural US imaging has been reported to overcome these limitations [12]. US/PET/CT fusion imaging facilitates endoscopic biopsy procedures in PET-positive abdominal lesions inconspicuous on morphologic imaging and it allows the biopsy to be performed in the most metabolically active region of the lesion [13].

The combination of these imaging modalities to enhance visualization of the tumor and to provide dynamic feedback would extend the standard of care for GI malignancy staging, diagnosis, and interventional therapies and could reduce the risks for adverse events and toxicity to surrounding vital abdominal structures. This is especially true when trying to assess treatment response from radiotherapy (RT) for pancreatic tumors where CT provides limited data and increased SUV on PET could be due to inflammation as opposed to residual disease. Recently, texture patterns and changes observed on CT before and after RT appear to correlate with the clinical outcomes of pancreatic ductal adenocarcinoma [14]. Correlation of these radiomic changes with US may provide additional information that could personalize treatment. Additionally, incorporation of US into the radiation treatment planning process can facilitate dose escalation of areas that appear to be more hypoxic or fibrotic. 

## 4. Technical Overview

Fusion of two imaging modalities requires a spatial co-registration to ensure that the real-time dynamic imaging data match the exact anatomy and spatial volume data of the reference imaging modality [1]. Achieving an accurate registration between images can be accomplished by defining multiple external or internal reference points in the patient. Internal points include common recognizable anatomic structures that are clearly visible on both images such as major arteries or specific points on an organ [1,3]. Registration points can include fiducial markers that have been deployed in the target lesion [1,2].

During the image fusion set up, two or more sets of co-registered images are co-displayed. This process can be done manually by an operator, who matches points and/or landmarks on both sets of images, or automatically by an algorithm, which matches pixels to voxels of two imaging datasets [15]. Semi-automated processes are also possible, which can either be conducted by constraining the registration with manual intervention or by manually adjusting the results of an initial registration [15].

Electromagnetic (EM) navigation is used to track the movement and position of the imaging devices within the region of the target lesion. This is accomplished by placing an electromagnetic field generator in close proximity to the patient and incorporating electromagnetic sensor coils within the interventional or sonographic imaging device [1]. These sensor coils are tracked in real time for the orientation and position of the US probe and are co-registered and re-formatted in a projection to match the previously acquired CT, MRI, or PET/CT images [1].

There are many challenges that present with image fusion technologies. In reality, many of the imaging data points collected are constantly being changed by tissue deformation caused by body movement such as respiration and gastrointestinal lumen contractions [1,3,16,17]. Additionally, a difference in body positioning during different modalities can affect the calibration of the image fusion. In order to avoid these significant deviations, the co-registration phase should occur in the same respiratory phase (when possible) as when the previously acquired images/data were taken. Further, the patient should ideally be in the same position (prone or supine) or an alpha cradle (immobilization used during radiation therapy) as when the CT or MRI was obtained when co-registering the data with US or EUS imaging. Co-registration involving image fusion with EUS can also be distorted due to the required pushing maneuvers of the echoendoscope against the gastric or duodenal wall for deep evaluation of the liver and pancreas [2]. The image fusion system is able to account for these slight distortions through another means called deformable image registration [3]. Deformable image registration allows the operator of image fusion to manually re-calibrate and modify images to account for the involuntary and voluntary movements, allowing fusion technology to be feasible in image-guided interventions. Some other partial limitations of the use of real-time image fusion include the high costs of the systems and the supplementary physician time needed for examination and image fusion itself.

A typical image fusion session combines the EM field generator, external physical markers, and the image fusion software in the following steps (Figure 1):Place the EM tracking and field generator system in close proximity to or under the patient and connect it to the computer running the fusion imaging (FI) software;Place one or more active marker disks on the patient’s xiphoid process;Place the EM sensor inside the navigation catheter in the working channel of the endoscope/echoendoscope;Load the pre-procedure CT scans in the FI software;Create a 3D model of the patient anatomy;Co-register the EUS patient space with the CT space;Identify and navigate towards the target using dual visualization of the EUS image and its corresponding virtual section through the CT volume;Make fine adjustments to the registration if necessary;Once the target is reached, the EUS is fixed in place; the navigation catheter is retracted and replaced with a FNA (fine needle aspiration) needle for biopsy collection.

## 5. Current Clinical Applications of Image Fusion in GI Malignancies

Regarding the role of image fusion in the management of GI malignancies, liver applications have been the most extensively investigated, either in experimental or commercially available systems. Real-time US fusion imaging of the liver is gaining more acceptance as it enables further diagnosis and interventional therapy of focal liver lesions that are difficult to visualize using conventional B-mode ultrasound (Table 1).

Studies have shown that image fusion allows for a better detection and characterization of focal liver lesions. The detection rate of small hepatocellular carcinomas (<3 cm) with US increased from 78.8% to 90.5% after using real-time US fused with CT or MR images [18]. Jung et al. reported a tendency towards increased detection of liver lesions with fusion of contrast-enhanced US and CT when compared to CT alone [5]. Furthermore, Rennert et al. [19] conducted a retrospective study evaluating 100 patients with benign or malignant liver lesions using image fusion scans of contrast-enhanced US (CE-US) with contrast-enhanced CT or MRI. In 12 patients, additional lesions were found using fusion imaging, resulting in a change in therapeutic strategy. This is in line with the results of other studies showing that liver lesions became more visible either on B-mode ultrasound or on CE-US when fused with previously obtained images (CT, MRI, or PET) [6]. Stang et al. showed that US-CT image fusion can improve the assessment of small liver lesions compared to “mentally fused” separate US and CT images, and this technique may be useful in staging patients with colorectal cancer as well [20].

Considering the results of these aforementioned studies, image fusion enables more accurate identification of hepatic tumors and, therefore, it can offer important advantages for targeting liver lesions during minimally invasive procedures such as biopsies and percutaneous ablations or for radiation treatment planning.

During the past two decades, imaging-guided tumor ablation using either chemical or thermal energy has emerged as one of the most effective loco-regional treatment modalities for small malignant hepatic tumors [21]. Among them, radiofrequency ablation (RFA) has been the most widely used method for the local treatment of small hepatocellular carcinomas (HCCs) and colorectal cancer liver metastasis due to its safety and effectiveness as well as a reasonably good clinical outcome, especially in lesions <3 cm in size [22,23,24,25]. However, an important limitation of percutaneous image-guided RFA is the absence of an ideal tool to guide and monitor the RFA procedure. CT is not able to provide real-time guidance and has a weakness in low-contrast hepatic tumors, while in US, there are several blind spots in the liver including the liver dome, the tip of the left lateral segment, and below the ribs. Furthermore, US has the limitation of ineffectively monitoring the procedure since gas clouds can interfere with the evaluation of the relationship between the index tumor and ablation zone [21].

The usefulness of real-time fusion imaging for RFA of hepatic tumors has been evaluated in clinical practice. Several studies have shown that image fusion techniques resulted in better feasibility of US-guided percutaneous RFA for HCCs with poor conspicuity on conventional B-mode US [26,27,28]. Moreover, image fusion in RFA can improve the operator’s confidence [29] and reduce the number of RFA sessions. Lee et al. [18] have shown that image fusion between the real-time working US and reference CT/MR images seems to be much more beneficial for small HCCs that are less than 2 cm in size, which may be invisible on a conventional US, especially in the background of advanced cirrhosis, compared to HCCs that are larger than 2 cm in size.

Image fusion can also help monitor RFA procedural adverse events. Potential complications and adjacent organ injury that arises after RFA might be avoided since fusion imaging can show the relationship between the ablation zone and vital structures, including the bile duct and large portal vein as well as adjacent organs.

Fusion imaging techniques can also accurately evaluate the ablation margins of RFA for liver metastasis or HCC. Usually, the ablative margin cannot be precisely evaluated on B-mode US and/or CE-US immediately after RFA because gas bubbles (due to the ablation) hide the tumor and the surroundings. The development of image fusion has made it possible to visualize the ablative margin on US. By an overlay of preoperative and postoperative US, the tumor image could be projected onto the white ablation zone in real time. Therefore, US–US overlay image fusion could show the ablative margin during the RFA procedure [30]. US–US image overlay fusion guidance can contribute to obtaining sufficient margins for RFA therapy. Recently, Li et al. showed that US-CT/MRI image fusion is also an accurate approach for evaluating ablative margins after tumor ablation based on both an in vitro model and in a clinical study [31].

Promising results were obtained for imaging guided microwave ablation of HCC undetectable by conventional ultrasound [32,33]. Another group tested a fusion system for contrast-enhanced ultrasound (CE-US) follow-up in HCC patients treated with transcatheter arterial chemoembolization (TACE). The method enabled a more exact mapping of the lesions and, thereby, a better evaluation of the residual tumor perfusion [34].

**Table 1 diagnostics-12-03218-t001:** Overview of the clinical applications of image fusion for liver tumors.

Reference	Year	Image Modalities for Fusion	No. of Patients	Clinical Application
Jung et al. [5]	2009	CE-US/CT/MRI	20	Assessment of the vascularization and perfusion of liver tumors
Rennert et al. [19]	2011	CE-US/CT/MRI	100	Localization and diagnosis of hepatic lesions in patients with primary hepatic cancer or liver metastases
Stang et al. [20]	2012	US/CT	64	Identification of hepatic metastases in patients with colorectal cancer
Song et al. [27,35]	2013	US/CT/MRI	120	Identification and ablation with RFA of hepatocellular carcinomas not visible on conventional US
Hakime et al. [26]	2017	US/CT	35	Targeting of liver metastases for percutaneous microwave ablation
Mauri et al. [28]	2014	US/CT/MRI	295	Targeting and thermal ablation of liver tumors undetectable with US alone
Minami et al. [25]	2014	US/CT/MRI	147	Guidance of RFA in hepatocellular carcinomas with poor conspicuity on B-mode US
Lee et al. [18]	2013	US/CT/MRI	137	Detection of small hepatocellular carcinomas for RFA
Minami et al. [30]	2016	US/US	10	Visualization of the ablative margin of RFA for liver metastases
Li et al. [31]	2017	CE-US/CT/MRI	24 (phantom models)	Evaluation of radiofrequency ablative margin
Liu et al. [32]	2012	US/CT/MRI	18	Real-time guidance of microwave ablation for hepatocellular carcinoma undetectable by conventional US
Zhang et al. [33]	2017	US/CT	19	Real-time three-dimensional guidance of percutaneous microwave ablation for hepatocellular carcinoma
Ross et al. [34]	2010	CE-US/CT/MRI	20	Evaluation of the results after transcatheter arterial chemoembolization for hepatocellular carcinoma

## 6. EUS Image Fusion—Is It Feasible?

EUS has provided gastroenterologists a tool to generate real-time high-resolution images of target organs for accurate diagnosis and staging of abdominal and thoracic malignancies. However, EUS is still not widely utilized by many gastroenterologists because of a difficult learning curve and challenging navigation techniques that accompany the small viewing plane [36,37]. There is a significant clinical impact for the use of EUS in the management of pancreaticobiliary and mediastinal diseases, especially for cytological/histological diagnosis and guidance of minimally invasive interventions [38,39,40].

EUS-guided interventions and staging are heavily dependent on the skills and experience of the endoscopists; therefore, fusion of CT, MRI, or PET/CT with EUS can provide more anatomical information to facilitate the navigation of the endoscope, as well as provide enhanced EUS evaluations due to better visualization of the target lesions, including evaluation of treatment response, and normal adjacent structures [2,41].

Image fusion with EUS uses the same technical preparation as mentioned previously, which includes an electromagnetic (EM) field generator with an EM tracking system, the fusion imaging navigation software, and a tracking electrode placed on the patient’s chest (xiphoid process). In addition, a miniature EM sensor attached to the end of the echoendoscope probe or a navigation catheter equipped with an EM sensor can replace the biopsy needle in the echoendoscope such that the EUS probe and suction can still function [1,2,42].

To date, clinical studies on EUS-guided image fusion have been limited, but promising results have been seen in two feasibility studies that analyzed efficacy, accuracy, image registration errors, total time to reach lesions, and precision of reaching the target lesion [2,42]. EUS–CT image fusion allowed for easier navigation and profiling of the target tumor and/or surrounding anatomical structure. The image fusion technique permitted direct side-by-side comparison of the target lesion, which allows for the multidisciplinary team to have a better visualization for further surgical or radiation treatment planning. Additionally, the complex vascularity of the abdomen surrounding the pancreas was better visualized for EUS-FNA, promoting greater patient safety and procedure time.

Obstein et al. [42] had noted a registration error of the CT image to the EUS image planes to be approximately 5 mm. Similarly, Gruionu et al. [2] had reported that the co-registration had to be manually re-aligned many times. Based on the endosonographer’s feedback, the EUS–CT imaging limitation did not add significant extra-procedural time, but there were multiple disruptions in order to manually realign the co-registrations of the imaging modalities. The studies concluded that EUS–CT image fusion is technically feasible and possibly lowered the learning curve for understanding and navigating EUS. Another added benefit of fusion imaging for EUS-guided procedures is the ability to compare pre-procedural CT images and the 3D reconstruction with the real-time manipulations that are performed under EUS guidance. Therefore, for procedures such as EUS-guided pancreatic cyst drainage or EUS-guided placement of fiducial markers, the real-time effect of the procedure can be dynamically compared with the pre-procedural anatomy to confirm the treatment outcome.

One additional limitation with EUS–CT imaging fusion was with the co-registration’s failure to accurately adjust for various physiological movements such as breathing, endoscopic manipulation, and patient movement [1,2,16,17,43]. Both case series cited minimal relative observed motion between the static CT and dynamic EUS images of approximately 3 mm [42]. During EUS, patients are normally in the left lateral decubitus position, which ensures that gravity causes minimal shifts in anatomical structures in the left upper quadrant. However, sedated patient respirations and endoscopic manipulation via transgastric/transduodenal imaging were seen to have negligible distortions in the visualization of the target anatomy, although the issue can be addressed by slightly updating the co-registration points based on nearby vascular structures [15,42].

Our group is currently developing a software platform (EUS/CT image fusion) with the following functions: automatic processing of pre-procedure CT for 3D rendering and segmentation; automatic nodule detection and procedure planning; automatic organ/tumor segmentation; automatic registration of patient’s CT for endoscopy procedures; localization and tracing of the position of therapeutic devices using electromagnetic or optical tracking technologies; virtual visualization of the medical instruments on the pre-procedure CT stack; augmented reality for virtual visualization of the patient’s anatomy over intraprocedural video. A prototype of the software has been tested on a custom-made pig organ model (Figure 2a). Initially, a CT of the model was performed using a pancreatic protocol. The next steps were: acquisition of the images, segmentation, 3D reconstruction, and uploading into IDEAR software. The visualization of the echoendoscope position in the 3D CT cube was possible. Co-registration of the EUS and CT images allowed for the same section to move in real time (Figure 2b). The combination of real-time EUS with CT enhanced the visualization of the targeted lesions/organs, allowing for the performance of complex therapeutic procedures. Moreover, it improved the operator’s confidence. Consequently, EUS would be less dependent on the operators’ skills, thus allowing for a widespread and uniform use of the technique.

## 7. Summary Points

Real-time US fusion imaging (CT/MR) allows for a better detection and characterization of focal liver lesions;Image fusion can offer important advantages for targeting liver lesions during minimally invasive procedures such as biopsies and percutaneous ablations or for radiation treatment planning;EUS–CT image fusion allows for easier navigation and profiling of the target tumor and/or surrounding anatomical structure;EUS–CT image fusion can lower the learning curve for understanding and navigating EUS.

## 8. Conclusions

Image fusion techniques encompassing multiple imaging modalities appear to be feasible and have been observed to increase visualization accuracy during interventional and diagnostic applications. The standard of care of GI malignancies can be enhanced by integrating image fusion so that better tumor diagnosis, staging, and multi-disciplinary treatment planning can be accomplished. These preliminary developments in the clinical application of image fusion involving CT/MRI/PET with EUS/US have shown great promise in facilitating diagnosis of small tumors and guidance of biopsies and interventional ablations. Further clinical research is needed to overcome current limitations before the widespread use of fusion imaging in managing GI malignancies can be achieved. Among them, the challenge of fusing images with large anatomical variation due to organ deformation should be the focus of future investigations to evaluate the benefit and accuracy of deformable image registration for this application. 

## Figures and Tables

**Figure 1 diagnostics-12-03218-f001:**
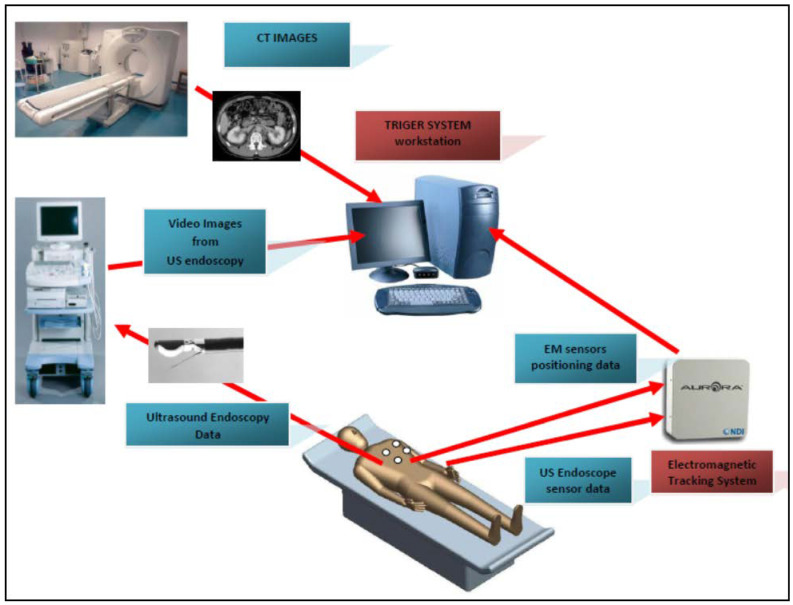
A typical FI procedure flow. The pre-procedure CT images are used to reconstruct a virtual 3D map of the patient. The FI software (Triger^TM^) is used to superimpose the EM sensor locations onto the 3D model. The live EUS images are integrated with the 3D model to find the clinical target.

**Figure 2 diagnostics-12-03218-f002:**
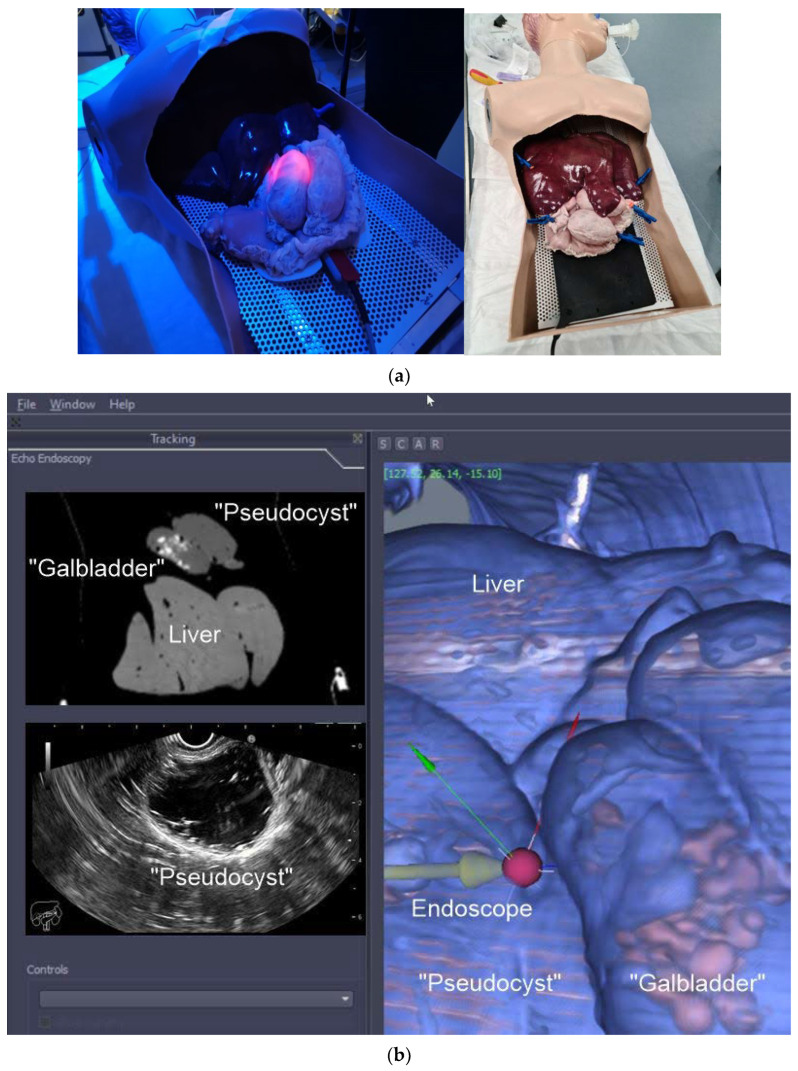
(**a**) Custom-made pig organ model. A custom-made pig organ model was used to test the image fusion software. The design of the model included: overtube; stomach; “pseudocyst”; “gallbladder” (filled with “stones”); liver; (**b**) Image fusion (EUS/CT) testing. EUS–CT fusion in real time showing the 3D reconstruction of the segmented CT with a phantom “gallbladder with stones inside”, “pseudocyst”, and liver (C). The oblique section shows the co-registered large-field CT, showing all 3 organs, and narrow EUS image, showing only the pseudocyst (A and B).

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
