# Peer review of "Image Fusion Involving Real-Time Transabdominal or Endoscopic Ultrasound for Gastrointestinal Malignancies: Review of Current and Future Applications"

_diagnostics, 2022, doi:10.3390/diagnostics12123218_

Round 1

Reviewer 1 Report (Previous Reviewer 2)

I have found that authors’ response to my comment is not satisfactory. Although the authors added Fig. 2, it is still difficult to realize the superiority of real time ultrasonography (US) with fusion of CT/MRI images. I strongly recommend the authors provide movie of US fused with CT/MRI to make this manuscript appropriate to this journal. Furthermore, I would like to point out that there are no figure captions of Fig. 2 and that it is unclear what a green arrow and a red circle indicate in Fig. 2b (C).

Another point:

1. (Line 120) (abdominal and EUS): This description is unclear.

Author Response

Reviewer 2 Report (Previous Reviewer 1)

All the corrections are included in the paper by the authors. There is no need for further review, and the paper may be accepted.

Author Response

This manuscript is a resubmission of an earlier submission. The following is a list of the peer review reports and author responses from that submission.

Round 1

Reviewer 1 Report

The abstract needs quantification. The methodology used in the paper is neither review /survey or exploration of new ideas. The section 1 and 2 needs more enhancement which misleads the readers. Section 5 to be explored with added points and tables. Section 6 is far away from the objective of the paper.

The conclusion needs to be modified. The presentation method is not as good as a scientific paper without a vigour.

Reviewer 2 Report

This review concisely summarizes the concept, methodology, and clinical application of image fusion in diagnosis and treatment. However, this review lacks impact because it emphasizes the superiority of image fusion only by text. Since this is a paper about imaging studies, more images, not just text, should be provided. Adding representative movies that merge real-time ultrasound with CT, MRI, and PET-CT images will help audience get strong impression.